# Clinicopathological Significances of Positive Surgical Resection Margin after Radical Prostatectomy for Prostatic Cancers: A Meta-Analysis

**DOI:** 10.3390/medicina58091251

**Published:** 2022-09-09

**Authors:** Minseok Kim, Daeseon Yoo, Jungsoo Pyo, Wonjin Cho

**Affiliations:** 1Department of Urology, Chosun University Hospital, Chosun University School of Medicine, Gwangju 61453, Korea; 2Department of Urology, Daejeon Eulji University Hospital, Eulji University School of Medicine, Daejeon 35233, Korea; 3Department of Pathology, Uijeongbu Eulji University Hospital, Eulji University School of Medicine, Uijeongbu 11759, Korea

**Keywords:** biochemical recurrence, meta-analysis, prostatic cancer, surgical margin

## Abstract

*Background and Objectives*: This study aims to elucidate the positive rate and the clinicopathological significance of surgical margin after radical prostatectomy (RP) through a meta-analysis. *Materials and Methods*: This meta-analysis finally used 59 studies, including the information about the positive surgical margin (PSM) and those clinicopathological significances after RP. The subgroup analysis for the estimated rates of PSM was evaluated based on types of surgery, grade groups, and pathological tumor (pT) stages. We compared the clinicopathological correlations between positive and negative surgical margins (NSM). *Results*: The estimated PSM rate was 25.3% after RP (95% confidence interval [CI] 21.9–29.0%). The PSM rates were 26.0% (95% CI 21.5–31.1%) 28.0% (95% CI 20.2–37.5%) in robot-assisted RP and nerve-sparing RP, respectively. The PSM rate was significantly higher in high-grade groups than in low-grade groups. In addition, the higher pT stage subgroup had a high PSM rate compared to the lower pT stage subgroups. Patients with PSM showed significantly high PSA levels, frequent lymphovascular invasion, lymph node metastasis, and extraprostatic extension. Biochemical recurrences (BCRs) were 28.5% (95% CI 21.4–36.9%) and 11.8% (95% CI 8.1–16.9%) in PSM and NSM subgroups, respectively. Patients with PSM showed worse BCR-free survival than those with NSM (hazard ratio 2.368, 95% CI 2.043–2.744%). *Conclusions*: Our results showed that PSM was significantly correlated with worse clinicopathological characteristics and biochemical recurrence-free survival. Among the results in preoperative evaluations, grade group and tumor stage are useful for the prediction of PSM.

## 1. Introduction

Prostate cancer was the most diagnosed cancer in men, and it was reported as the fourth most diagnosed cancer in the entire population [1]. Radical prostatectomy (RP) is one of the most effective and most used treatment methods for patients with localized prostate cancer. The surgical techniques have been developed from open RP in the 20th century to laparoscopic surgery and robotic surgery in recent decades [2,3]. There was no statistically significant difference in surgical, oncological, and functional results between these surgical techniques [4]. Despite advances in surgical procedures, 30% of patients undergoing RP still experience biochemical recurrence (BCR) [5]. In addition, in some cases in 20–30% of prostatic cancers progress to metastatic cancer and die [6]. A positive surgical margin (PSM) means that cancer cells are found at the surgical margin in the pathologic specimen after RP. The neurovascular bundle, bladder neck, and distal urethral sphincter are preserved to maintain urinary continence and erectile function, increasing the risk of PSM [7]. It has been reported that the operator’s surgical skills affect PSM or BCR in open or laparoscopic RP, but a recent large scale retrospective study showed that the surgical learning curve had no effect in robot-assisted radical prostatectomy [8,9]. The rate of PSM after RP was reported to occur at about 11 to 38% [10]. Previous studies have reported on the impacting factors causing BCR after RP, and among them, several studies have been published on the relevance of PSM. Although several studies have shown that patients with PSM after RP have a worse prognosis than those without PSM [11], some discrepancies in the clinicopathological significance of PSM are present. PSM is associated with BCR, prostate cancer survival rate, and distant metastasis [12,13]. However, some studies have reported that PSM is not significantly related to the patient’s oncological prognosis [14] and Dev, Harveer S., et al. reported that the length of PSM and apical PSM were related to BCR [15].

We performed this study to elucidate the positive rate and the clinicopathological significance of surgical margin after RP through a meta-analysis. In addition, a subgroup analysis, based on types of surgery, grade groups, and pathological tumor (pT) stages, was conducted in the present study.

## 2. Materials and Methods

### 2.1. Published Study Search and Selection Criteria

The literature search was performed using the PubMed and MEDLINE databases through 30 June 2020. The search was performed using the following keywords: “(prostate or prostatic) and (cancer or adenocarcinoma)” and “(radical prostatectomy)” and “(positive surgical margin).” The titles and abstracts of searched articles were primarily screened for exclusion. PICO (population, intervention, comparator, outcomes) was defined as, (1) population: patients with prostatic cancer; (2) intervention: RP; (3) comparator: the presence of PSM; and (4) outcomes: the rate of PSM and BCR-free survival. Literature or systematic review articles were also screened to find additional eligible studies. The inclusion and exclusion criteria were as follows: (1) studies for PSM after RP were included, and (2) non-original articles, such as case reports or review articles were excluded.

### 2.2. Data Extraction

For the meta-analysis, data were extracted in the eligible studies as follows [7,16,17,18,19,20,21,22,23,24,25,26,27,28,29,30,31,32,33,34,35,36,37,38,39,40,41,42,43,44,45,46,47,48,49,50,51,52,53,54,55,56,57,58,59,60,61,62,63,64,65,66,67,68,69,70,71,72,73]: the first author’s name, study location, study year, type of surgery, number of patients analyzed, patients’ age, prostate-specific antigen (PSA), and rates of lymphovascular invasion, perineural invasion, lymph node metastasis, and extraprostatic extension. In addition, biochemical-free recurrence and survival rate by the positivity of surgical margins were extracted from eligible studies. For the quantitative aggregation of survival results, the correlation between PSM and survival rate was analyzed according to the hazard ratio (HR), using one of three methods. In studies not reporting the HR or its confidence interval (CI), these variables were calculated from the presented data using the HR point estimate, log-rank statistic or its *p*-value, and the O-E statistic (the difference between the number of observed and expected events) or its variance. If those data were unavailable, the HR was estimated using the total number of events, the number of patients at risk in each group, and the log-rank statistic or its *p*-value. Finally, if the only useful data were in the form of graphical representations of survival distributions, survival rates were extracted at specified times to reconstruct the HR estimate and its variance under the assumption that patients were censored at a constant rate during the time intervals. The published survival curves were evaluated independently by two authors to reduce variability. The HRs were then combined into an overall HR using Peto’s method [74].

### 2.3. Statistical Analyses

To perform a meta-analysis, the Comprehensive Meta-Analysis software package was used (Biostat, Englewood, NJ, USA). The PSM rates after RP were investigated from overall cases. The PSM rates based on types of surgery were obtained and calculated through subgroup analysis. In addition, the estimated rates of PSM according to grade group and pT stages were investigated. We compared various characteristics, including age, PSA, lymphovascular invasion, perineural invasion, lymph node metastasis, extraprostatic extension, and biochemical recurrence between patients with PSM and NSM. In this meta-analysis, among fixed and random effect models, interpretation was made using the values of a random-effects model. Heterogeneity between eligible studies was assessed using Q and I^2^ statistics and presented using *p*-values. In addition, the sensitivity analysis was conducted to assess the heterogeneity of eligible studies and the impact of each study on the combined effect. Statistical significances between subgroups were evaluated through a meta-regression test. To consider the publication bias, Egger’s test was used. If significant publication bias was found, the fail-safe N and trim-fill tests were performed to confirm the degree of publication bias. *p*-value < 0.05 was considered significant.

## 3. Results

### 3.1. Selection and Characteristics of Studies

A total of 436 studies were identified in the database, searching for the meta-analysis. Finally, 59 studies were selected according to the inclusion and exclusion criteria. Among the searched studies, 300 studies were excluded due to a lack of sufficient information. In addition, 75 reports were excluded due to being articles in a language other than English (*n* = 39) and non-original articles (*n* = 36). Two remaining reports were excluded as they focused on other diseases (Figure 1). The characteristics of the eligible studies are shown in Table 1.

### 3.2. The Positive Surgical Margin Rates after Radical Prostatectomy

The PSM rates ranged from 6.2 to 71.5% in the eligible studies. The estimated rate of PSM after RP was 25.3% (95% CI 21.9–29.0%) (Table 2). The robot-assisted RP subgroup showed 26.0% (95% CI 21.5–31.1%) the PSM rate. The PSM rates were 28.0% (95% CI 20.2–37.5%) and 30.1% (95% CI 26.8–33.6%) in nerve-sparing and non-nerve-sparing subgroups, respectively. Cases with the intraoperative frozen section showed 19.3% (95% CI 12.2–29.1%). However, the PSM rate of cases without the intraoperative frozen section was 29.5% (95% CI 23.4–36.3%).

PSM rates were 10.0% (95% CI 6.8–14.6%), 17.6% (95% CI 9.3–30.8%), 24.1% (95% CI 11.9–42.8%), 21.6% (95% CI 19.7–38.8%), and 36.2% (95% CI 11.3–71.7%) in grade groups 1, 2, 3, 4, and 5, respectively (Table 3). In a subgroup analysis based on pT stage, PSM rates were 13.5% (95% CI 10.2–17.7%), 41.4% (95% CI 33.4–49.8%), and 65.1% (95% CI 32.6–87.8%) in pT2, pT3, and pT3 stages, respectively. The multifocality of PSM was estimated at 30.9% (95% CI 22.9–40.1%). The apical PSM rate was 28.9% (95% CI 23.1–35.5%) after RP.

### 3.3. Comparison of Clinicopathological Characteristics between PSM and NSM

Next, the clinicopathological characteristics were compared between PSM and NSM. The mean PSA levels of PSM and NSM were 9.190 (95% CI 8.284–10.095) and 7.360 (95% CI 6.927–7.793), respectively (Table 4). There was a significant difference in mean PSA level between PSM and NSM (*p* < 0.001 in a meta-regression test). In addition, the lymphovascular invasion was significantly higher in the PSM subgroup than in the NSM subgroup (36.8%, 95% CI 29.4–45.0% vs. 25.6%, 95% CI 23.1–28.3%). Rates of lymph node metastasis were 9.7% (95% CI 5.9–15.6%) and 2.3% (95% CI 1.1–4.7%) in PSM and NSM subgroups, respectively. Extraprostatic extension was more frequently found in the PSM subgroup than in the NSM subgroup (63.9%, 95% CI 52.0–74.3% vs. 23.2%, 95% CI 15.0–34.1%; *p* < 0.001 in a meta-regression test). However, there was no significant difference in the patient’s age and perineural invasion between PSM and NSM.

### 3.4. Comparison of Biochemical Recurrence and Biochemical Recurrence-Free Survival between PSM and NSM

Rates of biochemical recurrence (BCR) were 28.5% (95% CI 21.4–36.9%) and 11.8% (95% CI 8.1–16.9%) in the PSM and NSM subgroups, respectively. There was a significant difference in BCR between the PSM and NSM subgroups (*p* < 0.001 in a meta-regression test). Comparing BCR-free survival, the PSM subgroup had a worse BCR-free survival rate than the NSM subgroup (hazard ratio 2.368, 95% CI 2.043–2.744; Figure 2).

## 4. Discussion

RP is the most common treatment option for localized prostatic cancers [75,76]. Microscopic examination of the RP specimen is performed for the entire prostate, including Gleason’s score, tumor extension, and surgical resection margin. After RP specimens, the presence of PSM is an important factor in predicting BCR and BCR-free survival [7,24,27,29,31,32,35,37,38,40,59,66,68]. However, in localized prostate cancer with PSM, the management after RP remains controversial [62]. If PSM is highly suggestive in the preoperative evaluation, it will be useful in establishing a treatment strategy and a postoperative follow-up. Previous studies have reported the correlation between PSM and clinicopathological characteristics by evaluating patients who underwent RP [7,16,17,18,19,20,21,22,23,24,25,26,27,28,29,30,31,32,33,34,35,36,37,38,39,40,41,42,43,44,45,46,47,48,49,50,51,52,53,54,55,56,57,58,59,60,61,62,63,64,65,66,67,68,69,70,71,72,73]. However, the conclusive information from the individual study is not fully understood. The present study is a meta-analysis to investigate the correlation between PSM and clinicopathological characteristics and BCR-free survival after RP.

In the previous studies, the PSM rate after RP had a wide range and was approximately 20% [62]. Radiologic examination may be the most effective tool for predicting PSM among preoperative evaluations. However, the prediction of PSM in preoperative evaluations is limited in daily practice. In the present meta-analysis, the estimated PSM rate was 25.3% (95% CI 21.9–29.0%). This estimation resulted from simple integration. PSM was correlated with pT stage [77], BMI [78], serum PSA level [79], cancer percentage in biopsy specimens [80,81], prostate weight [82], and tumor volume [83]. In pathologic examination, the Gleason score is evaluated for the overall tumor, regardless of the tumor portion at PSM. So, the interpretation of the correlation between the grade group and PSM can be limited. Coelho et al. [84] suggested that the clinical stage was the only independent factor for predicting PSM. Although previous studies have reported the predicting factors of PSM, the integrative evaluation is limited by various populations and surgical methods. Therefore, a meta-analysis is more appropriate for obtaining detailed information. In addition, to obtain the detailed information, an additional subgroup analysis was needed. As expected, the PSM rate was significantly correlated with the higher grade group and pT stage in the present meta-analysis.

In the present study, clinicopathological characteristics were compared between patients with PSM and NSM. Patients with PSM had more frequent lymphovascular invasion than those with NSM. In addition, the rate of lymph node metastasis was significantly higher in the PSM subgroup than in the NSM subgroup. In cases with PSM, a precise microscopic examination is needed to detect the lymphovascular invasion because of hidden lymph nodes and distant metastasis. In addition, the comparison of BCR and BCR-free survival between PSM with and without lymphovascular invasion is needed. Porten et al. reported that tumor volume was associated with PSM [85]. Since tumor volume is associated with PSA level, the evaluation for the difference in PSA is needed. Patients with PSM had higher PSA levels than those with NSM (9.190 vs. 7.360). However, there were no significant differences in age and perineural invasion between PSM and NSM subgroups. These factors, age, and perineural invasion, are included in the characteristics of prostate cancer.

In eligible studies, BCR rates ranged from 10.7 to 46.0% in PC with PSM [21,24,28,29,37,38,53,67,73]. BCR was significantly correlated with the Gleason score, preoperative PSA, and pathologic stage [75]. BCR was significantly higher in cases with PSM than in cases with NSM (28.5 vs. 11.8%). In addition, patients with PSM showed a worse BCR-free survival than those with NSM (HR 2.368, 95% CI 2.043–2.744). Some report that there was no correlation between PSM and cancer-specific survival in long-term follow-up [86,87]. Chapin et al. reported that tumor location was not associated with BCR [75]. In our results, PSM at the apex was detected in 28.9% of overall PSM. However, the PSM rate could not be obtained by other tumor locations due to insufficient information. Further evaluation is needed on the impact of tumor location on BCR and BCR survival.

Recently, the application of robot-assisted RP has been increased in localized prostatic cancers. Previous studies have reported that PSM rates were low in robot-assisted RP specimens [62]. Robot-assisted RP showed a slightly low PSM rate compared to other RPs. However, there was no significant difference in PSM rate in a meta-regression test (*p* = 0.688). Nerve-sparing surgery can be applied to diminish complications after RP. However, because the neurovascular bundles are anatomically located adjacent to the prostate, the possibility of increasing PSM is present [56]. In the previous systematic review and meta-analysis, nerve-sparing surgery was not correlated with an increased risk of PSM in patients with pT2 tumors [88]. Interestingly, the risk of PSM increased in the pT3 stage with nerve-sparing surgery [88]. In the present study, the PSM rate was lower in the subgroup with nerve-sparing RP than in the subgroup without nerve-sparing RP (28.0 vs. 30.1%). This is probably because the non-nerve-sparing subgroup is more likely to have worse oncological factors such as tumor burden or high PSA levels, compared to the nerve-sparing subgroup. Therefore, despite the difference in the surgical method, it is believed that the PSM rate was lower in the nerve-sparing subgroup. We additionally performed a detailed analysis of the impact of the nerve-sparing technique on PSM based on the pT stage. However, unlike the previous study, there was no significant difference in PSM rate by application of the nerve-sparing technique in the same pT stage (data not shown). Theoretically, the impact of the intraoperative frozen section on reducing PSM rate is important. The rate of PSM was lower in the subgroup with the intraoperative frozen section than in the subgroup without the intraoperative frozen section (19.3 vs. 29.5%). However, a meta-regression test could not be performed due to an insufficient number of studies. Although the surgical resection margin is actually negative, PSM is detected by loss or cauterization of periprostatic tissue in the pathological examination. In addition, tumor locations, including lateral locations, can be considered.

This study has some limitations. First, the impact of the length of PSM could not be investigated due to insufficient information. The evaluation of the length of involved PSM is recommended in the pathological examination for RP specimens [89]. Second, an additional analysis for the correlation with tumor multifocality, location, and volume is needed.

## 5. Conclusions

PSM was significantly correlated with frequent lymphovascular invasion, lymph node metastasis, BCR, and BCR-free survival. Patients with higher grade group and pT stage showed frequent PSM. Grade group and tumor stage in preoperative evaluations can be useful for predicting PSM. In addition, evaluating PSM will help establish a careful strategy for RP and postoperative follow-up observation.

## Figures and Tables

**Figure 1 medicina-58-01251-f001:**
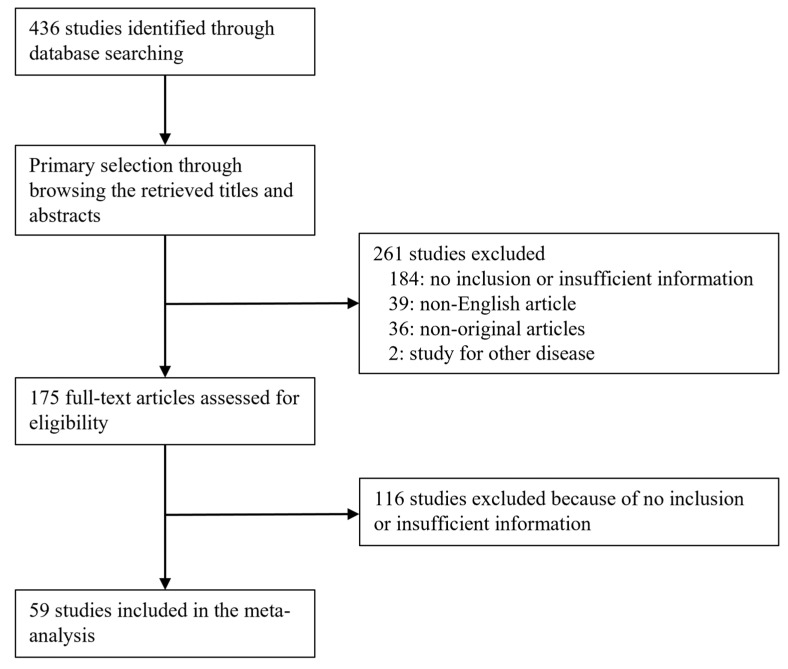
Flow chart.

**Figure 2 medicina-58-01251-f002:**
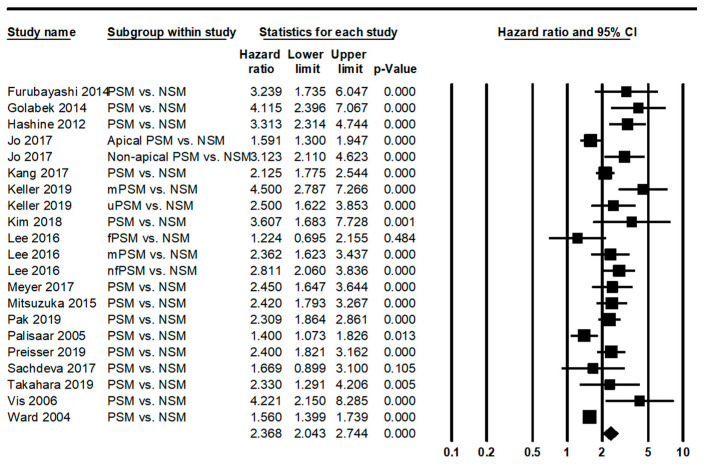
Forest plot for BCR-free survival between PSM and NSM (PSM: positive surgical margin; NSM: negative surgical margin; mPSM: multifocal PSM; uPSM: unifocal PSM; fPSM: focal PSM; nfPSM: non-single focal PSM) [7,24,26,27,29,31,32,35,37,38,40,41,48,53,59,66,68].

**Table 1 medicina-58-01251-t001:** Main characteristics of the eligible studies.

	Location	Operation	No of Patient			Location	Operation	No of Patients	
			Total	PSM				Total	PSM
Albisinni 2018 [16]	Belgium	Mixed	539	127	Poon 2000	USA	Non-robot	220	64
Aminsharifi 2019 [17]	USA	ND	4073	1490	Porcaro 2018	Italy	Mixed	476	327
Bianco 2003 [18]	USA	Non-robot	555	178	Poulakis 2006	Germany	Non-robot	182	31
Cangiano 1999 [19]	USA	Non-robot	301	72	Preisser 2019 (a)	Germany	Mixed	8770	579
Cannon 2005 [20]	USA	Non-robot	402	25	Preisser 2019 (b)	Germany	Mixed	346	*
Ceylan 2016 [21]	Turkey	Non-robot	130	93	Preston 2015	Canada	Mixed	6120	848
Dai 2019 [22]	China	Mixed	531	160	Rabbani 1998	USA	Non-robot	241	85
Eastham 2007 [23]	USA	Non-robot	2442	275	Rosen 1992	USA	Non-robot	144	33
Furubayashi 2014 [24]	Japan	ND	275	56	Sachdeva 2017	UK	Mixed	592	181
Ginzburg 2012 [25]	USA	Robot-assisted	1159	316	Salomon 2003	USA	Non-robot	371	66
Golabek 2014 [26]	Poland	Non-robot	295	86	Sayyid 2017	USA	Robot-assisted	200	48
Hashine 2012 [27]	Japan	Non-robot	505	194	Soeterik 2020	Netherlands	Robot-assisted	2574	844
Hollemans 2020 [28]	Netherlands	ND	835	284	Soulié 2001	France	Non-robot	212	71
Jo 2017 [29]	Korea	Robot-assisted	815	270	Stephenson 1997	USA	Non-robot	53	7
Jones 1990 [30]	Canada	Non-robot	199	92	Takahara 2019	Japan	Robot-assisted	230	52
Kang 2017 [31]	Korea	ND	1600	760	Tan 2019	USA, Puerto Rico	Mixed	45,426	4522
Keller 2019 [7]	Switzerland	Robot-assisted	973	315	Tatsugami 2017	Japan	Robot-assisted	3469	916
Kim 2018 [32]	Korea	Mixed	461	50	Tian 2019	China	Non-robot	418	142
Koizumi 2018 [33]	Japan	Mixed	450	64	Trabulsi 2009	USA	Robot-assisted	240	38
Konety 2004 [34]	USA	Non-robot	33	8	van den Ouden 1993	Netherlands	Non-robot	172	56
Lee 2016 [35]	Korea	ND	1733	473	Villers 2000	USA	ND	400	111
Menard 2008 [36]	France	Non-robot	640	180	Vis 2006	Netherlands	ND	281	66
Meyer 2017 [37]	Germany	ND	903	118	Volavšek 2018	ND	ND	107	29
Mitsuzuka 2015 [38]	Japan	Non-robot	1268	307	Ward 2004	USA	ND	7268	2103
Miyake 2010 [39]	Japan	Non-robot	127	14	Weldon 1995	USA	Non-robot	200	88
Pak 2019 [40]	Korea	ND	2013	404	Wu 2019	USA	ND	2796	476
Palisaar 2005 [41]	Germany	ND	1343	264	Yamada 2020	Japan	Robot-assisted	614	144
Park 2003 [42]	USA	Non-robot	221	43	Yu 2018	Korea	Mixed	3324	461
Park 2018 [43]	Korea	Mixed	546	179	Yuksel 2017	Turkey	Mixed	140	46
Partin 1993 [44]	USA	Non-robot	107	50					

No, number; PSM, positive surgical margin. *, included duplicated patients [7,16,17,18,19,20,21,22,23,24,25,26,27,28,29,30,31,32,33,34,35,36,37,38,39,40,41,42,43,44].

**Table 2 medicina-58-01251-t002:** The estimated rates of positive surgical margin after radical prostatectomy in prostatic cancers.

	Number of Subsets	Fixed Effect [95% CI, %]	Heterogeneity Test [*p*-Value]	Random Effect [95% CI, %]	Egger’s Test [*p*-Value]	Meta-Regression Test [*p*-Value]
Overall	58	20.2 [19.9, 20.5]	<0.001	25.3 [21,9, 29.0]	0.005	
Robot-assisted	12	26.9 [26.1, 27.7]	<0.001	26.0 [21.5, 31.1]	0.726	0.688 *
Others	28	26.5 [25.6, 27.4]	<0.001	27.2 [222, 32.7]	0.471	
Nerve-sparing	11	24.5 [23.7, 25.2]	<0.001	28.0 [202, 37.5]	0.661	0.580 ^†^
Non-nerve-sparing	8	29.4 [28.4, 30.5]	<0.001	30.1 [268, 33.6]	0.753	
Intraoperative frozen	2	19.1 [14.7, 24.5]	0.087	19.3 [122, 29.1]	0.077	-
Non-intraoperative frozen	1	29.5 [23.4, 36.3]	1.000	29.5 [234, 36.3]	-	

CI: confidence interval. * Comparison between robot-assisted and other radical prostatectomy. ^†^ Comparison between nerve-sparing and non-nerve-sparing radical prostatectomy.

**Table 3 medicina-58-01251-t003:** Detailed analysis of the estimated rates of positive surgical margin after radical prostatectomy in prostatic cancers.

	Number of Subsets	Fixed Effect [95% CI, %]	Heterogeneity Test [*p*-Value]	Random Effect [95% CI, %]	Egger’s Test [*p*-Value]	Meta-Regression Test [*p*-Value]
Grade group						
GG1 (Gleason score ≤6)	10	8.2 [7.8, 8.6]	<0.001	10.0 [6.8, 14.6]	0.535	Ref.
GG2 (Gleason score 3 + 4)	7	17.4 [16.1, 18.8]	<0.001	17.6 [9.3, 30.8]	0.918	0.100
GG3 (Gleason score 4 + 3)	6	22.7 [20.3, 25.3]	<0.001	24.1 [11.9, 42.8]	0.995	**0.010**
GG4/5 (Gleason score ≥8)	8	15.5 [14.4, 16.6]	<0.001	26.8 [16.8, 40.1]	0.078	**0.001**
GG4 (Gleason score 8)	4	13.0 [11.8, 14.3]	<0.001	21.6 [10.7, 38.8]	0.230	**0.036**
GG5 (Gleason score 9/10)	4	17.5 [15.0, 20.2]	<0.001	36.2 [11.3, 71.7]	0.307	**0.001**
pT stage						
pT2	17	13.8 [13.2, 14.5]	<0.001	13.5 [10.2, 17.7]	0.991	Ref.
pT3	15	34.5 [33.2, 35.7]	<0.001	41.4 [33.4, 49.8]	0.119	<**0.001**
pT4	3	65.1 [32.6, 87.8]	0.561	65.1 [32.6, 87.8]	0.064	**0.002**
Multifocal PSM rate	10	29.0 [27.3, 30.8]	<0.001	30.9 [22.9, 40.1]	0.370	
Apical PSM rate	11	25.0 [238.0, 263.0]	<0.001	289 [231.0, 355.0]	0.173	

CI: confidence interval; PSM: positive surgical margin; Ref: reference.

**Table 4 medicina-58-01251-t004:** Comparisons of clinicopathological parameters between positive and negative surgical margins after radical prostatectomy in prostatic cancers.

	Number of Subsets	Fixed Effect [95% CI]	Heterogeneity Test [*p*-Value]	Random Effect [95% CI]	Egger’s Test [*p*-Value]	Meta-Regression Test [*p*-Value]
Age (years)						
PSM	12	64.427 [64.341, 64.514]	<0.001	64.291 [63.149, 65.432]	0.787	0.970
NSM	9	63.492 [63.459, 63.523]	<0.001	64.273 [63.081, 65.465]	0.416	
PSA (ng/mL)						
PSM	10	8.368 [8.312, 8.425]	<0.001	9.190 [8.284, 10.095]	0.234	<**0.001**
NSM	8	6.867 [6.853, 6.881]	<0.001	7.360 [6.927, 7.793]	0.424	
Lymphovascular invasion (%)						
PSM	2	36.8 [29.4, 45.0]	0.470	36.8 [29.4, 45.0]	-	**0.005**
NSM	2	25.6 [23.1, 28.3]	0.710	25.6 [23.1, 28.3]	-	
Perineural invasion (%)						
PSM	3	24.5 [17.0, 33.9]	<0.001	41.2 [8.7, 83.7]	0.483	0.997
NSM	2	27.6 [20.8, 35.7]	<0.001	41.7 [4.5, 91.5]	-	
Lymph node metastasis (%)						
PSM	6	9.1 [7.4, 11.3]	0.001	9.7 [5.9, 15.6]	0.745	<**0.001**
NSM	6	3.8 [3.2, 4.6]	<0.001	2.3 [1.1, 4.7]	0.168	
Extraprostatic extension (%)						
PSM	3	61.7 [57.5, 65.7]	0.009	63.9 [52.0, 74.3]	0.413	<**0.001**
NSM	3	26.6 [25.0, 28.3]	<0.001	23.2 [15.0, 34.1]	0.620	
Biochemical recurrence (%)						
PSM	9	35.5 [32.9, 38.2]	<0.001	28.5 [21.4, 36.9]	0.034	<**0.001**
NSM	7	11.5 [10.4, 12.8]	<0.001	11.8 [8.1, 16.9]	0.991	

CI: confidence interval; PSM: positive surgical margin; NSM: negative surgical margin.

## Data Availability

The datasets generated during the current study are available from the corresponding author on reasonable request.

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
