# Peer review of "Clinicopathological Significances of Positive Surgical Resection Margin after Radical Prostatectomy for Prostatic Cancers: A Meta-Analysis"

_medicina, 2022, doi:10.3390/medicina58091251_

Round 1

Reviewer 1 Report

A meta-analysis evaluating the role of PSM after radical prostatectomy on oncological outcomes and BCR. 

I have some concerns regarding the work of the authors:

1. Introduction: lines 49-50: A recent article published on European Urology Oncology by Bravi et al. analyzing the impact of learning curve on over 8000 RARP, shows no impact of surgical experience on PSM and BCR. Please discuss it.

2. Introduction: lines 57. Only apical PSM have been linked to an higher risk of BCR (Dev H.S., et al. Surgical margin lenght and location affect recurrence rates after robotic prostatectomy. Urol Oncol. 2015. 33:107-113). Please discuss.

3. Materials and Methods: with did you only searched using PubMed and not other archives? Please discuss

4. Methods: no PICO reported, please amend

5. Please provide AMSTAR and SIGN checklist

6. Results: please express the rates as percentages. 

7. Lines 142-143: you wrote 2 times pT3

1. How can you explain the lower rates of PSM in nerve sparing procedures, considering the inherent risk of PSM carried with a closer dissection to the prostatic capsule in the this procedures, compared with non-NS RARP?

The findings are indeed interesting, even thought not new or unexpected. However the methodological concerns are strong and require modifications and improvements.

Author Response

Thank you very much for your kind advice. Thanks to your review, I can write a better paper.

We have revised the paper to actively reflect your advice.

I would like to ask you to review the paper again and look forward to good results.

Thank you again.

Reviewer 2 Report

1.      The authors need to check the spelling of the whole text. If an abbreviation is defined at the beginning, when it appears again the abbreviation should be used, e.g. radical prostatectomy in line 22.

2.      In lines 22-24, the authors wrote “The PSM rates were 0.260 (95% CI 0.215-0.311) and 0.193 (95% CI 0.122-0.291) in robot-assisted RP and nerve-sparing RP, respectively”. However, Table 2 shows that nerve-sparing RP were 0.280 (95% CI 0.202-0.375). The authors need to check that the data in the manuscript is consistent with the tables.

3.      The references cited in the introduction section are too old. It would have been more convincing if the authors had cited the most recent research in the field.

4.      The meaning of the sentence in line 49 is not clear enough. “It can affect the presence of extraprostatic capsular 49 extension and the operator's surgical skills” What does “It” refer to?

5.      The authors mentioned in the discussion section that Robot-assisted RP showed a slightly low PSM rate compared to other RP. There should be an increased analysis of why PSM rates were lower for robot-assisted RP and whether future improvements in this area will lead to lower PSM.

Author Response

(The authors gave the same response as above.)

Round 2

Reviewer 1 Report

The quality of manuscript has improved. No further comment on my side except the need for proofreading by a native english speaker.